# Calculation of Effective Thermal Conductivity for Human Skin Using the Fractal Monte Carlo Method

**DOI:** 10.3390/mi13030424

**Published:** 2022-03-10

**Authors:** Guillermo Rojas-Altamirano, René O. Vargas, Juan P. Escandón, Rubén Mil-Martínez, Alan Rojas-Montero

**Affiliations:** 1Departamento de Termofluidos, Instituto Politécnico Nacional, SEPI-ESIME Azcapotzalco, Av. de las Granjas No. 682, Col. Santa Catarina, Alcaldía Azcapotzalco, Ciudad de México 02250, Mexico; grojasa1400@alumno.ipn.mx (G.R.-A.); jescandon@ipn.mx (J.P.E.); alan_rojas102@hotmail.com (A.R.-M.); 2Escuela Militar de Ingenieros, Universidad del Ejército y Fuerza Aérea, Av. Industria Militar No. 261, Col. Lomas de San Isidro, Naucalpan de Juárez 53960, Mexico; rbnm2@hotmail.com

**Keywords:** effective thermal conductivity, fractal scaling, Monte Carlo, porous media, non-Newtonian fluid, power-law model, bioheat equation, human body

## Abstract

In this work, an effective thermal conductivity (ETC) for living tissues, which directly affects the energy transport process, is determined. The fractal scaling and Monte Carlo methods are used to describe the tissue as a porous medium, and blood is considered a Newtonian and non-Newtonian fluid for comparative and analytical purposes. The effect of the principal variables—such as fractal dimensions $D_{T}$ and $D_{f}$, porosity, and the power-law index, *n*—on the temperature profiles as a function of time and tissue depth, for one- and three-layer tissues, besides temperature distribution, are presented. ETC was improved by considering high tissue porosity, low tortuosity, and shear-thinning fluids. In three-layer tissues with different porosities, perfusion with a non-Newtonian fluid contributes to the understanding of the heat transfer process in some parts of the human body.

## 1. Introduction

The skin is the largest single organ of the body, enabling protection from the surrounding environment. It consists of several layers and plays an important role in thermoregulation, sensory, and host defense functions [1,2,3]. The skin is generally described by a three-layer tissue: epidermis, dermis, and hypodermis (also called subcutaneous) [4,5,6]. The thickness of these layers varies depending on the location of the skin. The epidermis is the outer layer (75–150 μm), this layer plays a barrier role between environment and organism [7,8]. The dermis is much thicker than the epidermis, in this, there are blood vessels, nerves, lymph vessels, and skin appendages. Dermis performances important functions in thermoregulation and supports the vascular network to supply the non-vascularized epidermis with nutrients. This layer is formed by an irregular network with wavy and unaligned collagen fiber bundles, allowing considerable deformations in all directions. The hypodermis is composed of loose fatty connective tissue. It is not part of the skin, but appears as a deep extension of the dermis, and depends on the age, sex, race, endocrine, and nutritional status of the individual [7,8,9,10]. The thermoregulation function of skin is realized mainly by modifying the blood flow, which is located in a microcirculatory bed, composed of arterioles, arterial and vein capillaries, and venules (blood perfusion). Blood perfusion has great effect on the heat transfer process in living tissues [8,11,12]. Heat transfer in human tissues takes place through different mechanisms, such as heat conduction, blood perfusion, metabolic heat generation, and external interactions [13,14]. One of the earliest models of heat transfer in biological tissues was developed by Pennes in 1948 [15], who proposed a model to describe the effects of metabolism and blood perfusion on the energy balance within tissue, this model is based on the classical Fourier’s law [15]. Wulff [16] questioned the assumptions of the Pennes model and provided an alternative analysis. He assumed that heat transfer between blood flow and tissue should be modeled proportionally to the temperature difference between these two media and not between the two temperatures of the blood flow. Klinger [17] consider the convective heat transfer caused by the blood flow inside the tissue, since this term was neglected by Pennes. Chen and Holmes [18] assumed that the total tissue control volume is composed of the solid-tissue subvolume and blood subvolume. They determined an effective thermal conductivity using the tissue porosity and the local mean tissue temperature, together with a simplified volume-averaging technique for the solid and tissue spaces. Weinbaum et al. [19,20] determined an effective thermal conductivity (ETC) based on the hypothesis that small arteries and veins are parallel and the flow direction is countercurrent, which is a function of the blood flow rate and vascular geometry. The model included a perfusion bleed-off term that apparently resembles the Pennes perfusion term. Weinbaum and Jiji [11] derived a simplified equation to study the influence of the blood flow on the tissue temperature distribution defining an ETC.

In biological tissues, many body parts reveal anisotropy in heat transport that can not be explained by the Fourier’s law [6,8]. This leads to formulation of thermal wave bioheat model based on two main approaches: the Maxwell–Cattaneo approach with heat flux time lag (also known as the single-phase approach), and the double-phase-lag (DPL) approach with relaxations in both the heat flux and temperature gradient propagation [21,22,23]. Various researchers have contributed in this area with analytical and experimental work. Hobiny and Abbas [24] presented an analytical solution of the hyperbolic bioheat equation under intense moving heat source. Alzahrani and Abbas [25] also presented an analytical approach, experimental temperature data, and a time sequential concept to obtain the thermal damage and temperature in a living tissue due to laser irradiation. Hobiny and Abbas [26] provided a method to determine numerical solutions for thermal damage of cylindrical living tissues using hyperbolic bioheat model. Hobiny et al. [27] proposed a new interpretation to study thermal damage in a skin tissue caused by laser irradiation, using the fractional order bioheat model. Hobiny et al. [28] presented an analytical method and experimental verification, to estimate thermal damage and temperature due to laser irradiation, using skin surface measurement data. Kumari and Singh [29] generated a space-fractional mathematical model of bioheat transfer to graphically analyze thermal behavior within living tissue, using a three-phase-lag constitutive relation. Li et al. [30] developed a generalized model of bioheat transfer to explore heat transport properties involving different thermal phase lagging effect. Important reviews and articles that the reader can consult additionally in this context are the following: classical mathematical models of bioheat [13]; developments in modeling heat transfer in blood perfused tissues [9,31]; bioheat models based on the porous media theory [32]; general heat transfer review [33,34]; concepts, derivation, and experimental versus porous media modeling [35]; and modeling and scaling of the bioheat equation [3].

Alternatively to continuum models, concepts that consider the tissue matrix, arteries, veins, and capillary vessels in a porous medium with specific porosity variations, ETC, and heat dispersion by blood flow have been developed [3,12,13,32,36]. Porous medium is defined as a material volume consisting of solid matrix with an interconnected pores. It is characterized by porosity, ratio of the pore space to the total volume of the medium, permeability, and tortuosity [32]. Khanafer and Vafai [36] remarked that the most appropriate treatment for heat transfer in biological tissues is the porous media theory because of fewer assumptions as compared with other models. Roetzel and Xuan [37] introduced a two-equation bioheat model in which the biological system is a porous media. It is divided into two different regions, the vascular and extravascular, without considering local thermal equilibrium between the two phases, introducing an equivalent ETC in the energy equations of blood and tissue [37]. Nakayama and Kuwahara [38] developed a model that consists of two energy equations based on the volume average theory (VAT), these equations are correct for all cases of thermal non-equilibrium.

The ETC is one of the most important thermo-physical properties for quantifying conductive heat transfer of porous media with gas, liquid, and solid phases [39]. The prediction of the ETC of porous media is essential to many engineering applications, such as thermally enhanced oil recovery, geothermal energy, and chemical and biological engineering [40]. With the development of computer technology, many numerical methods, such as Monte Carlo [41] and lattice Boltzmann [42], have been proposed to study the conductive heat transfer and evaluate the ETC of porous media. In addition to conventional methods based on Euclidean geometry, fractal geometry has been shown with evident advantages for addressing the complexity and multiple scales of porous media [43]. Hence, the fractal geometry has been successfully applied to characterize structures of transport processes in porous media [44,45]. Kou et al. proposed a fractal model for ETC of porous media based on fractal scaling law for water and gas phases in the pores [46]. Extensive studies have shown that most natural porous media and some synthetic porous media possess self-similar fractal scaling laws over multiple scales [45]. Therefore, two kinds of fractal models based on pore and solid phases have been proposed [47]. Fractal scaling laws can be applied to characterize the geometrical and morphological structures for pore and solid phases in porous media, respectively.

The fractal Monte Carlo method has been applied in different areas to model a porous media. Yu et al. [48] performed Monte Carlo simulations to predict the permeability of fractal porous media, their results were verified by comparison with the analytical solution for the permeability of bi-dispersed porous media. Zou et al. [49] used the Monte Carlo simulation technique to model the surface topography in a scale-invariant manner with the fractal nature of rough surfaces. Yu [44] presented a review article summarizing the theories, methods, mathematical models, achievements, and open questions in the area of flow in fractal porous media by applying the theory and technique of fractal geometry. Feng et al. [50] combined the Monte Carlo technique with fractal geometry theory to predict the thermal conductivity of nanofluids. Xu et al. [51] performed Monte Carlo simulations of radial seepage flow in the fractured porous medium, where the fractal probability model was applied to characterize the fracture size distribution. Vadapalli et al. [52] proposed a permeability estimation method for a sandstone reservoir, which considers the fractal behavior of pore size distribution and tortuosity of capillary pathways using Monte Carlo simulations. Xu et al. [53] used fractal Monte Carlo simulations to predict the effective thermal conductivity of porous media. Xiao et al. [54] employed the fractal Monte Carlo to simulate the Kozeny–Carman constant of fibrous porous media with the micropore size characterized by the fractal scaling law. Yang et al. [55] performed Monte Carlo simulations based on the fractal probability law to understand gas flow mechanisms and predict the apparent gas permeability of shale reservoirs.

In this work, the calculation of ETC for human skin using the fractal scaling and Monte Carlo methods is presented. This ETC involves a bundle of tortuous capillaries whose size distribution follow fractal scaling laws. The power-law model was chosen because of its simplicity in describing different non-Newtonian fluids by modifying a single parameter, in the case of blood as a shear-thinning fluid. The heat transfer process in the perfused tissue is analyzed, and a heat source is applied on the tissue surface for a period of time without reaching the degradation temperature. The tissue is considered as a uniform porous medium of one and three layers, which can be assigned different porosity and conductivity. The temperature profiles, as well as their distribution when modifying the main variables of the model, are presented. To the authors’ knowledge, there are no studies that determine an ETC for heat transfer in biological tissues, considering the techniques mentioned above and especially for non-Newtonian fluids.

## 2. Heat Transfer in Human Skin

Figure 1 presents the human skin structure, considering three layers; epidermis, dermis and hypodermis. These layers differ in having their own physical properties such as density, specific heat, thermal conductivity and porosity.

The first equation that described heat transfer in human tissue and included the effects of blood flow on tissue temperature on a continuum basis was presented by Pennes [15]. He presented the heat transfer analysis in the human forearm, considering the metabolic heat rate in the tissue and the perfusion heat source term. This last term has been the focus of attention since its inception and many researches have found alternative representations of the effect of blood perfusion on tissue heat transfer. The Pennes equation is given by:(1)$$\rho_{t}c_{t}\frac{\partial T_{t}}{\partial t}=\nabla \cdot \left(k_{t}\nabla T_{t}\right)+\rho_{b}c_{b}\omega_{b}\left(T_{a}-T_{t}\right)+q_{m},$$
where ρ, *c*, *T*, *t*, *k*, $q_{m}$, and $\omega_{b}$ are density, specific heat, temperature, time, thermal conductivity, metabolic heat production, and blood perfusion rate per unit volume of tissue, respectively. The subscripts *t*, *b*, and *a* refer to tissue, blood, and artery, respectively [15].

However, some inconsistencies in the Pennes model include the following: the thermal equilibrium take place in arteries and veins (not in the capillaries, as it assumes); it does not take into account any vascular architecture; and the most critical assumption is on the blood perfusion term, which is not a global term—it is local along the capillary and depends on direction.

## 3. Mathematical Modeling

Hyperthermia treatment consists of applying heat in a specific area of the human body. In this work, an external heat source is applied to an area of the forearm, as shown in Figure 2a. In addition, in Figure 2b, it is described that *H* is the total thickness of the tissue and the thicknesses of each layer are $H_{E}=0.04H$, $H_{D}=0.48H$, and $H_{H}=0.48H$. Initially, the tissue is at a constant temperature, $T_{c}∼$ 37 ${}^{\circ }$C, subsequently, a heat source is applied to an area of the tissue. The area around the heat application area is open to the surroundings, and generally is a temperature lower than the human body. For this work, the surrounding temperature is considered to be $T_{\infty }∼$ 25 ${}^{\circ }$C. Furthermore, Figure 2b shows a mathematical representation of the hyperthermia treatment, note that the deepest internal temperature is maintained at the body temperature. According to Weinbaum and Jiji [11], the simplified two-dimensional governing equation of heat transport in biological tissue is given by:(2)$${\left(\rho C_{p}\right)}_{t}\frac{\partial T}{\partial t}=\frac{\partial }{\partial x}\left(k_{eff}\frac{\partial T}{\partial x}\right)+\frac{\partial }{\partial y}\left(k_{eff}\frac{\partial T}{\partial y}\right)+q_{m},$$
where $k_{eff}$ is the effective thermal conductivity (ETC) and $q_{m}$ is the metabolic heat.

According to physical model showed in Figure 2, Equation (Equation 2) is subject to the following initial and boundary conditions:(3)$$\begin{array}{cc} T\left(x,y,0\right) & =T_{c}, \end{array}$$(4)$$\begin{array}{cc} \frac{\partial }{\partial x}T\left(0,y,t\right) & =0, \end{array}$$(5)$$\begin{array}{cc} \frac{\partial }{\partial x}T\left(W,y,t\right) & =0, \end{array}$$(6)$$\begin{array}{cc} T\left(x,0,t\right) & =T_{c}, \end{array}$$(7)$$\begin{array}{cc} -k_{t}\frac{\partial }{\partial y}T\left(x,H,t\right) & =h\left[T_{\infty }-T\left(x,H,t\right)\right],\;\frac{2}{5}W\geq x\geq \frac{3}{5}W, \end{array}$$(8)$$\begin{array}{cc} -k_{t}\frac{\partial }{\partial y}T\left(x,H,t\right) & =f,\;\frac{2}{5}W<x>\frac{3}{5}W,\\ f & =\left\{\begin{array}{cc} q_{app}=200\left[\frac{W}{m^{2}}\right], & t\leq t_{app},\\ h\left[T_{\infty }-T\left(x,H,t\right)\right], & t>t_{app}, \end{array}\right. \end{array}$$
where *W* and *H* are the width and the height of the domain, respectively. The *h*, $q_{app}$, and $t_{app}$ are the heat transfer coefficient, applied external heat, and application time, respectively.

ETC is an important parameter in Equation (Equation 2). Weinbaum and Jiji [11] proposed a vascular function $V\left(y\right)$ that can be constructed knowing the vascular data, which is the distribution of the arteries, veins, and capillaries. According to Weinbaum and Jiji [11], the vascular function increases with tissue depth [14]. In this work, the *representative elementary volume* (REV), is defined and the fractal scaling method is used to depict the vascular tissue structure.

### 3.1. Fractal Scaling Method

We consider a cubic REV as shown in Figure 3, with defined length side $L_{0}$. All REV capillaries extend throughout the volume from one side to the other as is showed in Figure 3.

The fractal scaling method establishes the relationship between the number and pore size in the porous medium. The fundamental fractal scaling law is applied to REV cross-section, as follows [56]:(9)$$N\left(>\lambda \right)={\left(\frac{\lambda_{\max}}{\lambda }\right)}^{D_{f}},$$
where *N*, $D_{f}$, λ, and $\lambda_{\max}$ are the number of capillaries, the fractal dimension, the equivalent diameter, and the maximum equivalent diameter of the capillaries in the REV cross-section, respectively. The number of capillaries with equivalent diameter between λ+dλ in the REV cross-section is:(10)$$-dN\left(\lambda \right)=D_{f}\lambda_{\max}^{D_{f}}\lambda^{-D_{f}-1}d\lambda .$$

The total number of pores in the range from $\lambda_{\min}$ to $\lambda_{\max}$ is obtained using Equation (Equation 9), as follows:(11)$$N_{T}\left(\leq \lambda_{\min}\right)={\left(\frac{\lambda_{\max}}{\lambda_{\max}}\right)}^{D_{f}},$$
where $N_{T}$ is the total pores. Dividing Equation (Equation 10) by Equation (Equation 11), we obtain:(12)$$-\frac{dN}{N_{T}}=D_{f}\lambda_{\min}^{D_{f}}\lambda^{-\left(D_{f}+1\right)}d\lambda =f\left(\lambda \right)d\lambda ,$$
where $f\left(\lambda \right)$ is the probability density function, which satisfies that $f\left(\lambda \right)\geq 0$. According to probability theory, $f\left(\lambda \right)$ must satisfy the following normalization relation or total cumulative probability:(13)$$-\int_{\lambda_{\min}}^{\lambda_{\max}}\frac{dN}{N_{T}}=\int_{\lambda_{\min}}^{\lambda_{\max}}f\left(\lambda \right)d\lambda =1-{\left(\frac{\lambda_{\min}}{\lambda_{\max}}\right)}^{D_{f}}\equiv 1.$$

The integration result of Equation (Equation 13) shows that it holds if—and only if—the following holds:(14)$${\left(\frac{\lambda_{\min}}{\lambda_{\max}}\right)}^{D_{f}}≅0.$$

The above equation implies that $\lambda_{\min}\ll \lambda_{\max}$, it must be satisfied for fractal analysis of a porous media. According to Yu et al. [48], Equation (Equation 14) can be considered as a criterion whether a porous medium can be characterized by fractal theory and technique. If Equation (Equation 13) is expressed as:(15)$$R\left(\lambda \right)=\int_{\lambda_{\min}}^{\lambda }f\left(\lambda \right)d\lambda =1-{\left(\frac{\lambda_{\min}}{\lambda }\right)}^{D_{f}},$$
then the pore diameter λ can be found as:(16)$$\lambda =\frac{\lambda_{\min}}{{\left(1-R\right)}^{1/D_{f}}}=\left(\frac{\lambda_{\min}}{\lambda_{\max}}\right)\frac{\lambda_{\max}}{{\left(1-R\right)}^{1/D_{f}}}.$$

On the other hand, the fractal path of the tortuous capillary can be described as follows [57]:(17)$$L\left(\lambda \right)=L_{0}^{D_{T}}\lambda^{1-D_{T}},$$
where $L\left(\lambda \right)$, $L_{0}$, and $D_{T}$ are capillary tortuous length, characteristic length of a straight capillary (REV side length), and fractal dimension describing the capillary tortuous length, respectively. An important parameter involved in the ETC is the total pore area, $A_{c}$, determined by Wu and Yu [58]:(18)$$A_{c}=-\int_{\lambda_{\min}}^{\lambda_{\max}}\frac{\pi \lambda^{2}}{4}dN\left(\lambda \right)=\frac{\pi D_{f}\lambda_{\max}^{D_{f}}\left(\lambda_{\max}^{2-D_{f}}-\lambda_{\min}^{2-D_{f}}\right)}{4\left(2-D_{f}\right)},$$
the total cross area, $A_{T}$, is:(19)$$A_{T}=L_{0}^{2}=\frac{A_{c}}{\phi },$$
where ϕ is the porosity.

### 3.2. The Fractal ETC of the REV

In order to obtain the ETC and according to Fourier’s law, the total heat flux in the REV is given by:(20)$$Q_{T}=k_{eff}A_{T}\frac{\Delta T}{L_{0}},$$
where $k_{eff}$ is the ETC including the tissue conductivity and convective blood flow in the capillaries, and ΔT is the difference temperature between two faces in the REV. The heat flux in a single tortuous capillary of the REV is:(21)$$Q_{c}=k_{b}\frac{\pi \lambda^{2}}{4}\frac{\Delta T}{L\left(\lambda \right)}=k_{b}\frac{\pi \lambda^{2}}{4}\frac{\Delta T}{L_{0}^{D_{T}}\lambda^{1-D_{T}}},$$
where $k_{b}$ is the blood thermal conductivity. The heat flux corresponding only to the tissue is expressed by:(22)$$Q_{t}=k_{t}A_{t}\frac{\Delta T}{L_{0}}=k_{t}\left(1-\phi \right)A_{T}\frac{\Delta T}{L_{0}},$$
where the subscript *t* refers to the tissue. According to superposition theorem, the total heat flux $Q_{T}$ is the addition of the fluxes as follows:(23)$$Q_{T}=Q_{t}+Q_{c}.$$

Substituting Equations (Equation 21) and (Equation 22) into Equation (Equation 23) the following equation is obtained:(24)$$k_{eff}=k_{t}\left(1-\phi \right)+k_{b}\frac{\phi \left(2-D_{f}\right)\lambda^{D_{T}+1}}{L_{0}^{D_{T}-1}D_{f}\lambda_{\max}^{D_{f}}\left(\lambda_{\max}^{2-D_{f}}-\lambda_{\min}^{2-D_{f}}\right)},$$
where $L_{0}$ is determined from Equation (Equation 19). According to the work presented by Weinbaum and Jiji [11], from an energy conservation balance in a countercurrent artery and mean value theory, derived the effective enhancement of the tissue conductivity. Therefore, by comparing the Weinbaum and Jiji ETC to Equation (Equation 24), it can be seen that the two equations have a similar form, proposing the following the expression:(25)$$k_{eff}=k_{t}\left(1+\Gamma \left(r\right)\psi \left(r\right)\right),$$
where Γ is a parameter that depends on conduction or convective heat transport [59], ψ is related to skin structure (also called dimensionless vascular geometry function) [14], and r is the position vector. This work focuses on the convective transport of blood. Hence, the dimensionless ETC for this work is given by:(26)$$k_{eff}^{*}=\frac{k_{eff}}{k_{t}}=\left[\left(1-\phi \right)+{\mathrm{Pe}}^{2}{\left(\frac{k_{b}}{k_{t}}\right)}^{2}\frac{\phi \left(2-D_{f}\right)\lambda^{D_{T}+1}}{L_{0}^{D_{T}-1}D_{f}\lambda_{\max}^{D_{f}}\left(\lambda_{\max}^{2-D_{f}}-\lambda_{\min}^{2-D_{f}}\right)}\right],$$
where $k_{eff}^{*}$ is the dimensionless ETC and Pe is the Peclet number defined as Pe = PrRe = $\rho_{b}C_{b}\lambda u/k_{b}$; where $\rho_{b}$, $C_{b}$, and *u* are the density, specific heat, and average blood velocity in the capillary, respectively.

### 3.3. Fractal Dimensions, $D_{f}$ and $D_{T}$

Equation (Equation 26) is a function of porosity, fractal dimensions, maximum and minimum diameters, conductivity ratio, and Peclet number, the latter being a function of physical properties, velocity, and diameter.

There is a relationship between porosity and fractal dimension, $D_{f}$, according to Yu and Li [56], is given by:(27)$$\phi ={\left(\frac{\lambda_{\min}}{\lambda_{\max}}\right)}^{D_{E}-D_{f}},$$
where $D_{E}$ is the Euclidean dimension ($D_{E}=2$ and 3, for two- and three-dimensional space, respectively). Another important aspect of Equation (Equation 27), from experimentation, it has been found that the ratio of minimum and maximum diameter, $\lambda_{\min}/\lambda_{\max}$, in several natural porous media, is the order of $10^{-2}∼10^{-4}$,[60]]. Feng et al. derived a generalized model covering a wide range of porosities, for the effective thermal conductivity, based on the fact that statistical self-similarity exists in porous media [61].

In this work, $D_{T}$ is established manually, taking into account whether $D_{T}>1$ the tortuosity is present and $D_{T}=1$ are straight capillaries.

### 3.4. Non-Newtonian Fractal Velocity

The Peclet number in Equation (Equation 26) is defined by:(28)$$\mathrm{Pe}=\mathrm{PrRe}=\frac{\rho_{b}C_{b}}{k_{b}}\lambda u,$$
where the velocity *u* is a function of the microcapillary diameter, is obtained from the non-Newtonian fluid flow in a single microcapillary, as presented by Zhang [62], as follows:(29)$$q_{c}={\left[\frac{dp}{dL_{0}}\frac{L_{0}^{1-D_{T}}2^{D_{T}-1}}{\mu_{b}\left(D_{T}+1\right)D_{T}}\right]}^{\frac{1}{n}}\frac{n\pi }{D_{T}+3n}{\left(\frac{\lambda }{2}\right)}^{\frac{D_{T}}{n}+3},$$
where $q_{c}$, *p*, and $\mu_{b}$ are the flow rate in a single microcapillary, pressure, and dynamic blood viscosity, respectively. By considering $q_{c}=uA_{sc}$, where $A_{sc}$ is the area of a single capillary and the velocity can be determined as:(30)$$u={\left[\frac{dp}{dL_{0}}\frac{L_{0}^{1-D_{T}}2^{D_{T}-1}}{\mu \left(D_{T}+1\right)D_{T}}\right]}^{\frac{1}{n}}\frac{n}{D_{T}+3n}{\left(\frac{\lambda }{2}\right)}^{\frac{D_{T}}{n}+1},$$
where *n* is the power-law index of the constitutive equation, when n=1 the Newtonian velocity is recovered. For n<1 and n>1 describes shear-thinning and shear-thickening fluid behavior, respectively. Figure 4 presents the average velocity and Peclet number as a function of the capillary diameter for different values of the power-law index, *n*, considering two values of porosity, ϕ=0.1 and ϕ=0.5. By increasing the pore diameters, the flow through the tissue is greater for all cases. This effect is magnified when the power-law index, *n*, decreases, which indicates a lower resistance to flow, because the viscosity decreases, which is characteristic of shear-thinning fluids. The opposite case can be seen in this figure for shear-thickening fluids (n>1). The number of Peclet is proportional to the velocity, therefore they have the same tendency. On the other hand, by increasing the porosity, both *u* and Pe have a slight increase because the pores are very small. Figure 5 shows the average velocity and Peclet number as a function of the fractal tortuosity, $D_{T}$, for different values of porosity, ϕ, considering two values of power-law index, n=1 and n=0.6. For this case, the maximum pore diameter was taken into account. Keeping constant porosity and increasing tortuosity, $D_{T}$, the velocity and Peclet number both decrease. This is correct for complex vascular architectures where the flow experiences higher resistance as well as being very small. For straight or slightly tortuous capillaries—where the highest velocities and Peclet numbers are found—there is a large increase as the power-law index decrease, as shown in Figure 5.

### 3.5. Monte Carlo Method

The Monte Carlo method is used to establish the pore size and the porous medium distribution, as is shown in Figure 6. Figure 6a,b present one- and three-layer tissue with different porosities, with ϕ=0.1 (one layer) and ϕ=0, ϕ=0.5, and ϕ=0.05 for epidermis, dermis, and hypodermis, respectively. Yu et al. [48] was the first to propose the fractal Monte Carlo methodology to simulate the transports in fractal porous media. In Figure 6c for one-layer tissue and Figure 6d for three-layer tissue show the Monte Carlo simulations, which are performed in the range of $\lambda_{\min}$–$\lambda_{\max}$. Figure 6d shows the pore sizes variation for dermis and hypodermis layers, since the porosity of the epidermal layer is not considered. The pore size variation is determined using the Monte Carlo method in Equation (Equation 16), as follows:(31)$$\lambda_{i}=\frac{\lambda_{\min}}{{\left(1-R_{i}\right)}^{1/D_{f}}}=\left(\frac{\lambda_{\min}}{\lambda_{\max}}\right)\frac{\lambda_{\max}}{{\left(1-R_{i}\right)}^{1/D_{f}}},$$
where $\lambda_{i}$ is the variation of the pore diameter and $R_{i}$ are the random numbers between 0−1. On the other hand, the random numbers also help us to construct the porous medium presented in Figure 6. Equation (Equation 31) is derived from Equation (Equation 9), which implies that there is only one largest pore in the REV cross-section [44]. This is consistent to the pore size distribution shown in Figure 6.

### 3.6. Dimensionless Governing Equation

The dimensionless variables are:(32)$$\bar{x}=\frac{x}{H};\;\bar{y}=\frac{y}{H};\;\tau =t\frac{\alpha_{t}}{H^{2}};\;\theta =\frac{T-T_{\infty }}{T_{c}-T_{\infty }},$$
where $\alpha_{t}$ is the tissue thermal diffusivity. Substituting dimensionless variables of Equation (Equation 32) into Equation (Equation 2), the dimensionless governing equation is:(33)$$\frac{\partial \theta }{\partial \tau }=\frac{\partial }{\partial \bar{x}}\left(k_{eff}^{*}\frac{\partial \theta }{\partial \bar{x}}\right)+\frac{\partial }{\partial \bar{y}}\left(k_{eff}^{*}\frac{\partial \theta }{\partial \bar{y}}\right)+\Phi_{m},$$
where $k_{eff}^{*}=k_{eff}/k_{t}$ is defined in Equation (Equation 26) and $\Phi_{m}=q_{m}H^{2}/k_{t}\left(T_{c}-T_{\infty }\right)$ is the dimensionless metabolic heat generation. Respectively, the dimensionless initial and boundary conditions are given by: (34)$$\begin{array}{cc} \theta \left(\bar{x},\bar{y},0\right) & =1, \end{array}$$(35)$$\begin{array}{cc} \frac{\partial }{\partial \bar{x}}\theta \left(0,\bar{y},\tau \right) & =0, \end{array}$$(36)$$\begin{array}{cc} \frac{\partial }{\partial \bar{x}}\theta \left(2,\bar{y},\tau \right) & =0, \end{array}$$(37)$$\begin{array}{cc} \theta \left(\bar{x},0,\tau \right) & =1, \end{array}$$(38)$$\begin{array}{cc} \frac{\partial }{\partial \bar{y}}\theta \left(\bar{x},1,\tau \right) & =Bi\left[\theta_{\infty }-\theta \left(\bar{x},1,\tau \right)\right],\;\frac{2}{5}\geq \bar{x}\geq \frac{3}{5}, \end{array}$$(39)$$\begin{array}{cc} \frac{\partial }{\partial \bar{y}}\theta \left(\bar{x},1,\tau \right) & =f^{*},\;\frac{2}{5}<\bar{x}>\frac{3}{5},\\ f^{*} & =\left\{\begin{array}{cc} \frac{H}{k_{t}\Delta T}q_{\mathrm{app}}, & \tau \leq \tau_{\mathrm{app}},\\ Bi\left[\theta_{\infty }-\theta \left(\bar{x},1,\tau \right)\right], & \tau >\tau_{\mathrm{app}}, \end{array}\right. \end{array}$$
where *Bi* is the Biot number defined as *Bi* = $hH/k_{t}$, $\theta_{\infty }$ is the dimensionless temperature of the environment, and $f^{*}$ takes the boundary condition according to the dimensionless time of application $\tau_{\mathrm{app}}$.

## 4. Results

The numerical results presented in this section were generated from a numerical code developed in the Fortran programming language. Equation (Equation 33) subject to boundary conditions Equations (Equation 34)–() was solved using an explicit finite difference method.

The values used for the tissue physical properties are: $\rho_{t}=1200$ kg/m${}^{3}$, $C_{p,t}=3600$ J/kg$\cdot^{\circ }$C, $k_{t}=0.293$ W/m$\cdot^{\circ }$C. When considering three-layer tissue, their corresponding thermal conductivities are: $k_{e}=0.25$ W/m$\cdot^{\circ }$C, $k_{d}=0.45$ W/m$\cdot^{\circ }$C, and $k_{h}=0.2$ W/m$\cdot^{\circ }$C, for epidermis, dermis, and hypodermis, respectively. The blood physical properties are: $\rho_{b}=1052$ kg/m${}^{3}$, $C_{p,b}=3800$ J/kg$\cdot^{\circ }$C, and $k_{b}=0.582$ W/m$\cdot^{\circ }$C [2,63]. The metabolic heat is $q_{m}=368.1$ W/m${}^{3}$[63]. The heat transfer coefficient due to convection and radiation in surroundings is: h=5 W/m${}^{2}\cdot^{\circ }$C. The environmental temperature was chosen as $T_{\infty }=25$${}^{\circ }$C. For the three-layer model, the main changes in porosity are considered to be in the dermis, due to the greater interaction between the tissue and the vascular network [8]. This work does not take into account thermal degradation (thermal damage) in tissue, which occurs at 44 ${}^{\circ }$C and higher [64].

### 4.1. ETC Analysis

According to Equation (Equation 26), ETC depends on the conductivity ratio, porosity, ϕ, fractal coefficients, $D_{f}$, $D_{T}$, and Peclet number. In this work, both conductivities of tissue and blood are constant; therefore, the conductivity ratio is also constant. Once the porosity value is assigned manually, $D_{f}$ can be determined using Equation (Equation 27). The Peclet number depends on the physical properties of the blood, capillary diameter, and velocity, see Equation (Equation 28). An important contribution of this work is that it allows the analysis of blood as a Newtonian and non-Newtonian fluid. This is possible through the power-law model, which is immersed in the velocity calculation Equation (Equation 30). This equation also depends on the fractal coefficients, in addition to the viscosity and the power-law index, *n*. The latter makes it possible to consider a Newtonian fluid (n=1) and non-Newtonian fluids such as shear-thinning fluid (n<1) and shear-thickening n<1. The blood is a shear-thinning fluid described by a power-law index value of n=0.6, according to Johnston et al. [65].

Figure 7a presents the ETC as a function of the fractal coefficient $D_{T}$ for different porosity values, in addition for two values of the power-law index n=1 and n=0.6. $D_{T}$ describes the capillary tortuous length, the influence of $D_{T}$ on the ETC is in the range of 1–1.3, which is in accordance with that reported by Yu and Cheng [57]. By increasing the porosity, the ETC also increases, due to higher blood flow through the tissue having a higher thermal conductivity. An important increase is found in the ETC when considering blood as a non-Newtonian fluid, since it increases with porosity as mentioned above, but especially for a shear-thinning fluid such as blood (n=0.6).

Figure 7b shows the fractal dimension as a function of porosity. When porosity tends to unity, the fractal dimension tends to two. That for a surface indicates that it is totally occupied by pores, which corresponds to a fractal dimension of two, it is consistent according to Yu et al. [66].

Figure 8 exhibits the ETC as a function of fractal dimension $D_{f}$, and the porosity ϕ; for two pore diameter values, $\lambda =4\times 10^{-4}$ and $\lambda_{\max}=5\times 10^{-4}$, and by considering straight capillaries $D_{T}=1$ for different values of the power-law index. $k_{eff}^{*}$ has an increase by augmenting the pore diameter, but the main improvement is for shear-thinning fluids, due to the lower resistance that the fluid experiences in large pores, increasing the velocity and generating a higher energy dissipation.

### 4.2. Code Validation

To validate the ETC, the solution of the Weinbaum and Jiji [11] (WJ) is compared with the present work using a one-layer tissue with porosity ϕ=0.05, straight capillaries $D_{T}=1$, $k_{b}/k_{t}=1.9863$, $D_{f}$ is determined by Equation (Equation 27) and Newtonian fluid n=1. The Peclet number depends on the pore diameter λ and the velocity *u*, WJ determined Pe=20, which is in accordance with this work of Pe=16.3 for maximum pore size. Figure 9 shows the comparison of the temperature profile as a function on time and a function of deep-tissue layer (at τ=0.09) between WJ model and the present work. The ETC obtained by Weinbaum and Jiji [11] is as follows:(40)$$\frac{k_{eff,WJ}}{k_{t}}=\left[1+Pe_{0}^{2}V\left(\bar{y}\right)\right],$$
where $Pe_{0}=2\rho_{b}c_{b}a_{0}u_{0}/k_{b}=20$, $V\left(\bar{y}\right)=A+B\bar{y}+C{\bar{y}}^{2}$ with $A=6.32\times 10^{-5}$, $B=-15.9\times 10^{-5}$ and $C=10\times 10^{-5}$. The metabolic heat $\Phi_{m}=q_{m}L^{2}/k_{t}\left(T_{c}-T_{\infty }\right)=0.094$.

The comparison between WJ and the present work exhibited in Figure 9, for the temperature as a function of (a) time and (b) deep of the tissue layer, both fit in a good agreement. The comparison between the temperature distribution of the WJ model (Equation 40) and the present work (Equation 26) is shown in Figure 10. The heat source is applied on the central part of the tissue surface, as shown in Figure 2. The temperature distribution corresponds to a heat source application dimensionless time of τ=0.09. The temperature distribution of the WJ model shows a higher evolution, indicating a higher heat transfer in the tissue. The difference between models according to temperature contours is less than 2%.

### 4.3. Dynamical Test Simulations

#### 4.3.1. One-Layer Tissue Analysis

To evaluate the different parameters involved in the ETC, a dynamic test is performed, which consists of applying a heat source on the tissue surface for a period of time, removing the source when the thermal relaxation process begins, the scheme and conditions are shown in Figure 2. This test does not reach the tissue degradation temperature, T< 44 ${}^{\circ }$C [8,64].

Figure 11 presents the temperature profile as a function of time and tissue layer depth for one-layer (at τ=0.09), by modifying the fractal coefficient $D_{T}$ in the range of 1–1.1, with porosity ϕ=0.1 and Newtonian fluid n=1. There are no significant changes when the heat source is applied, at the end of the relaxation process is when there is a difference between the WJ model and this work. This difference is due to the fact that there is a better energy transfer process in the present model, but there is no appreciable effect of $D_{T}$ due to the low tissue porosity.

Figure 12 shows the temperature profile as a function of time and tissue layer depth (at τ=0.09) for one-layer tissue by modifying the porosity ϕ in the range of 0.01 to 0.5, with $D_{T}=1.05$ and Newtonian fluid n=1. As the porosity increases, higher temperatures are reached when the heat source is applied, and lower temperatures are reached at the end of the relaxation process. This is due to increased perfusion in the tissue, on the other hand, blood has a higher thermal conductivity which improves heat transfer in the porous medium.

Figure 13 and Figure 14 exhibit the temperature profile as a function of time and tissue layer depth (at τ=0.09) one layer for two porosity values ϕ=0.1 and ϕ=0.5, respectively. Modifying the power-law index *n* in the range of 0.6 to 1 and $D_{T}=1.05$. In the case of porosity ϕ=0.1, there are no significant changes when modifying the power-law index, due to low perfusion through the tissue. For porosity ϕ=0.5, blood perfusion is increased and the effect of the power-law index is reflected both in the application of the heat source and in the thermal relaxation process. For a shear-thinning fluid n=0.6, there is a greater dissipation of energy and therefore it reaches lower temperatures compared with a Newtonian fluid, in addition to experiencing a higher thermal relaxation due to a greater influence of the convective condition of the surface, as shown in Figure 14a. The effect of higher porosity is also reflected in the temperature profile as a function of tissue depth, as it presents distortions, mainly for the non-Newtonian fluid with n=0.6, see Figure 14b.

#### 4.3.2. Three-Layer Tissue Analysis

The structure of the skin is very complex and is generally considered in three layers—epidermis, dermis, and hypodermis—as is shown in Figure 1. The thickness of these layers varies depending on the location of the skin and many factors, such as age, sex, race, endocrine, and nutritional status of the individual [7,8,9,10].

Figure 15 shows the temperature profile as a function of time and tissue three-layer depth (at τ=0.09), each layer with its own conductivity, $k_{e}=0.25$ W/m$\cdot^{\circ }$C, $k_{d}=0.45$ W/m$\cdot^{\circ }$C and $k_{h}=0.2$ W/m$\cdot^{\circ }$C, for epidermis, dermis, and hypodermis, respectively, [8]. The porosity ϕ varies uniformly throughout the tissue in the range of 0.01–0.5, with $D_{T}=1.05$ and non-Newtonian fluid n=0.6. The shear-thinning effect increases the heat transfer in tissue, which is reflected in the temperatures reached when applying the heat source and in the thermal relaxation process. The effect of having three different conductivities can be seen clearly in the temperature profile as a function of tissue depth, where changes in the slope of the curve at the layer interfaces are presented, as is shown in Figure 15b. This behavior has been reported by Xu et al. [2].

Figure 16 shows the temperature profile comparison between one- and three-layer tissue at different depths as a function of time and tissue depth, the properties of the three-layer tissue are those used in Figure 15. In the case of a one-layer tissue, the uniform porosity is ϕ=0.5, for the case of three-layer tissue the following porosities are assigned ϕ=0, 0.5, and 0.05 for epidermis, dermis, and hypodermis, respectively. These values were assigned according to the fact that the epidermis has very low porosity, the dermis contains the highest density of capillaries, and the hypodermis is composed of loose fatty connective tissue. The other variable values are $D_{T}=1.05$ and n=0.6 [8]. Considering blood as a non-Newtonian shear-thinning fluid, in addition to different tissue properties, generates important modifications in the temperature profiles both in time and depth. In contrast, uniform properties throughout the biological tissue, as shown in Figure 16a,b.

Figure 16a shows red dots on temperature profiles used to indicate the time corresponding to the instantaneous temperature distribution presented in Figure 17. This figure shows the comparison of temperature distribution between the one- and three-layer tissues at different times. In the case of three-layer tissues, the effect of the different tissue properties on the temperature contours, which show small variations due to porosity depending on the area. In the case of one-layer tissues, the heat transfer is more similar to homogeneous solids, which have higher thermal conductivity at the top and bottom of the tissue compared with the three-layer model. Finally, the right half is overlapped on the left half of the temperature distribution for one- and three-layer tissue to demonstrate the asymmetry generated by the global effect of all the variables involved in the model, as shown in Figure 18.

## 5. Conclusions

In this work, the ETC for human skin using the fractal scaling and Monte Carlo methods was obtained. These methods were used to describe the tissue as a porous medium, the blood was considered as Newtonian and non-Newtonian shear-thinning fluid. The numerical code developed was validated by comparing the ETC obtained by Weinbaum and Jiji [11] and the present work using one-layer tissue with the same porosity, straight capillaries, and Newtonian fluid. The difference between models according to temperature contours is less than 2%.

The ETC involves various parameters, such as fractal dimensions $D_{T}$ and $D_{f}$, porosity, and the power-law index *n*; in order to evaluate these parameters, dynamical tests were performed, which consisted of applying a heat source on the tissue surface for a period of time—when the heat source was removed, the thermal relaxation process began. The effect of the main parameters on the temperature profiles as a function of time and tissue depth, for one- and three-layer tissue, besides temperature distribution, were presented. The main findings of this work are the following:The effect of fractal dimension $D_{T}$ on the ETC was mainly in the range of 1–1.3.Higher porosity improves ETC, due to increased blood flow through the tissue, having a higher thermal conductivity.In one-layer tissues of low porosity, no significant changes in ETC were found. Increasing porosity, the effect of the power-law index is reflected in both heating and relaxation processes.The Peclet number increases substantially due to the combination of large pore diameters and shear thinning fluids.In three-layer tissues with different porosities, perfusion with a shear-thinning fluid contributes to the understanding of the heat transfer process in some parts of the human body.The ETC involves the main variables of the heat transfer process in human skin; moreover, it is easy to implement for other case studies.

## Figures and Tables

**Figure 1 micromachines-13-00424-f001:**
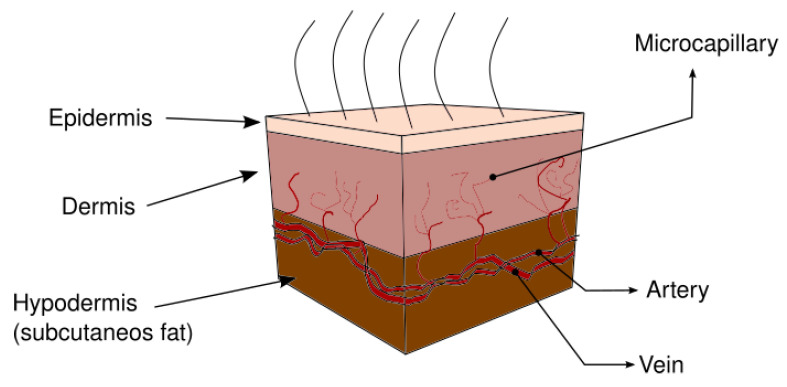
Human skin structure.

**Figure 2 micromachines-13-00424-f002:**
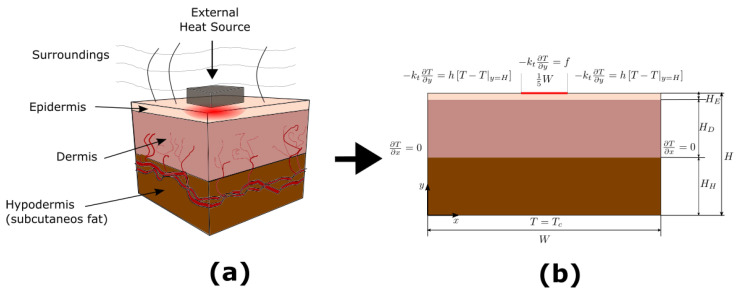
(**a**) Hyperthermia treatment; (**b**) human skin three-layer model, and the corresponding boundary conditions.

**Figure 3 micromachines-13-00424-f003:**
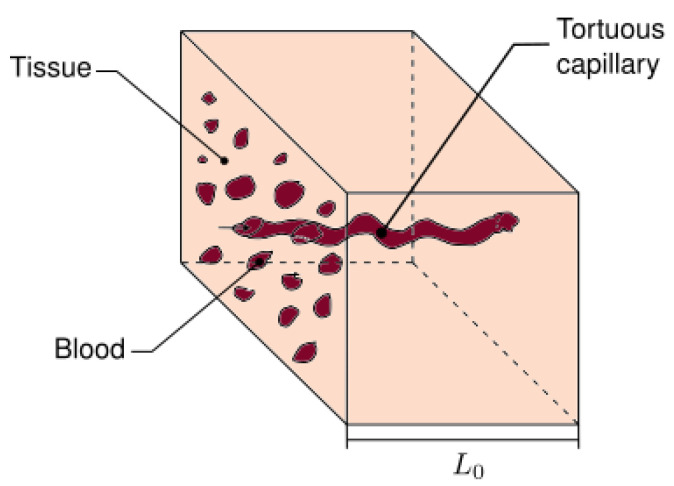
Representative elementary volume of human skin.

**Figure 4 micromachines-13-00424-f004:**
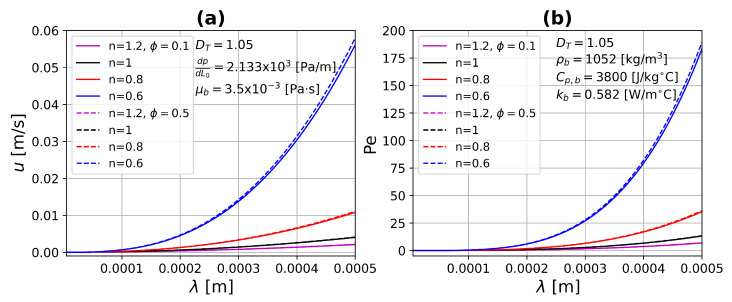
(**a**) Average velocity and (**b**) Peclet number as a function of capillary diameter for different values of the power-law index *n*. In a single microcapillary—continuous lines (ϕ=0.1), and dashed lines (ϕ=0.5).

**Figure 5 micromachines-13-00424-f005:**
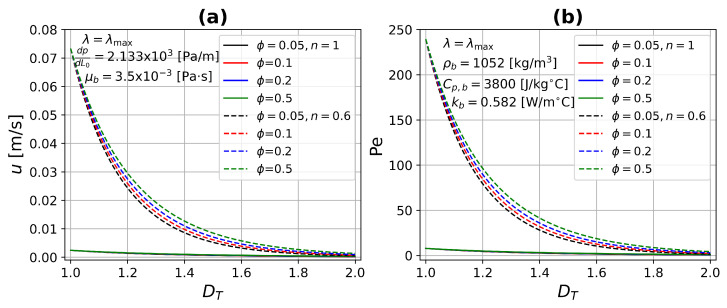
(**a**) Average velocity and (**b**) Peclet number as a function of fractal tortuosity $D_{T}$ for different porosity values. For both in a single microcapillary, continuous lines (Newtonian fluid), and dashed lines (non-Newtonian fluid).

**Figure 6 micromachines-13-00424-f006:**
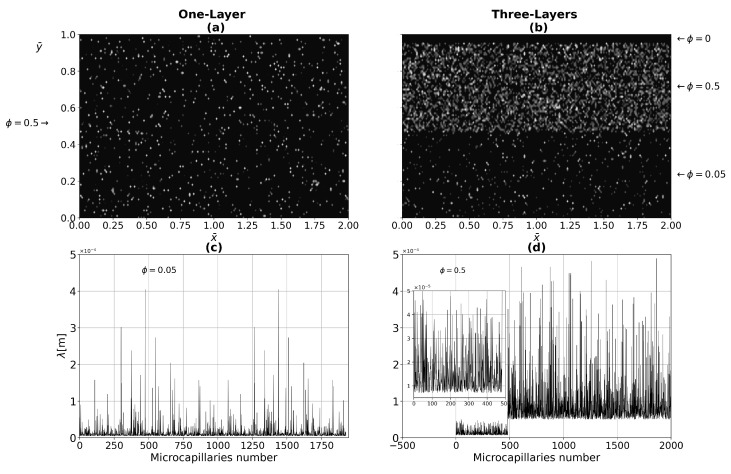
Two-dimensional pore size distribution for (**a**) one-layer tissue with ϕ=0.1, (**b**) three-layer tissue with ϕ=0,0.5, and 0.05 for epidermis, dermis, and hypodermis, respectively. (**c**,**d**) The simulated pore sizes by the Monte-Carlo technique in the range of $\lambda_{\min}=5\times 10^{-6}$ to $\lambda_{\max}=5\times 10^{-4}$.

**Figure 7 micromachines-13-00424-f007:**
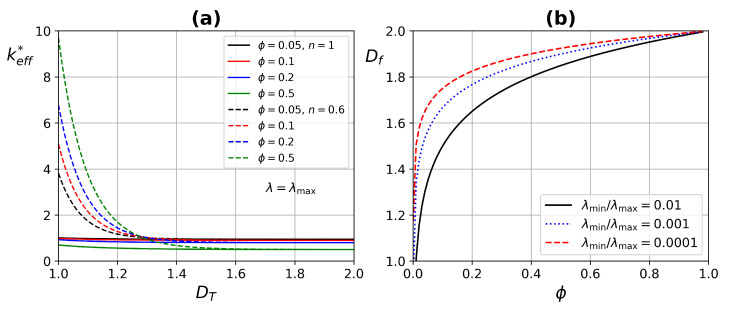
(**a**) Effective thermal conductivity as a function of fractal dimension $D_{T}$. Continuous lines (Newtonian fluid) and dashed lines (non-Newtonian fluid). (**b**) Fractal dimension $D_{f}$ as a function of porosity, for different ratios $\lambda_{\min}/\lambda_{\max}$.

**Figure 8 micromachines-13-00424-f008:**
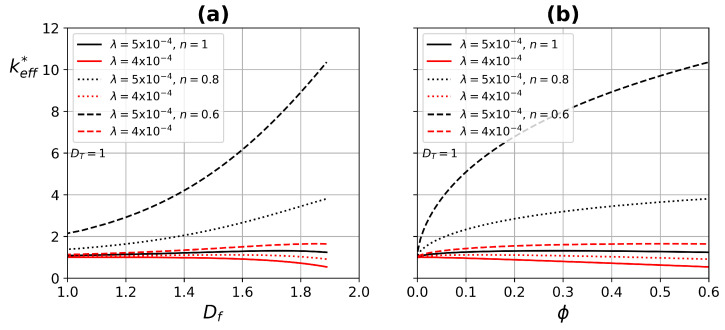
Effective thermal conductivity as a function of (**a**) fractal dimension $D_{f}$, and (**b**) porosity ϕ. Continuous lines (Newtonian fluid) and dashed lines (non-Newtonian fluid).

**Figure 9 micromachines-13-00424-f009:**
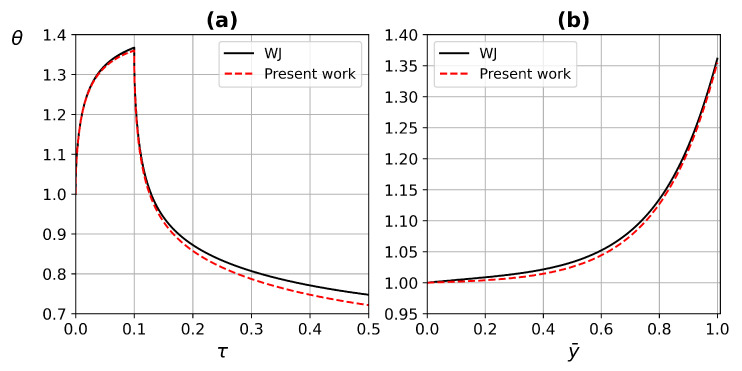
Comparison of temperature profile between WJ and the present work as a function of (**a**) time, and (**b**) tissue layer depth.

**Figure 10 micromachines-13-00424-f010:**
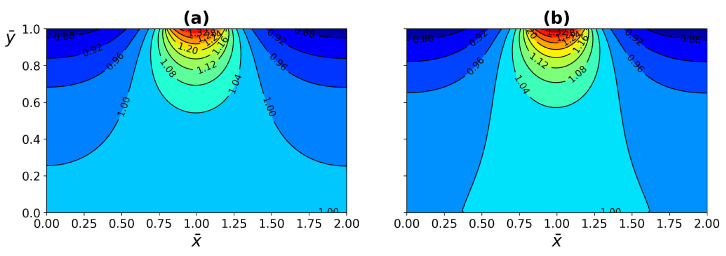
Comparison of temperature distribution between (**a**) WJ model, and (**b**) present work at dimensionless time. Just before the heat source was removed.

**Figure 11 micromachines-13-00424-f011:**
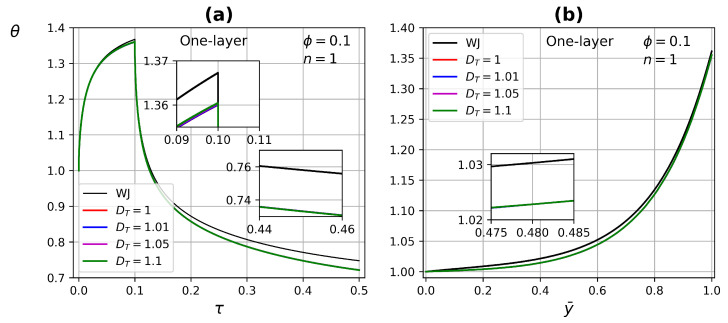
Temperature profiles as a function of (**a**) time and (**b**) tissue layer depth, for different $D_{T}$ values at τ=0.09.

**Figure 12 micromachines-13-00424-f012:**
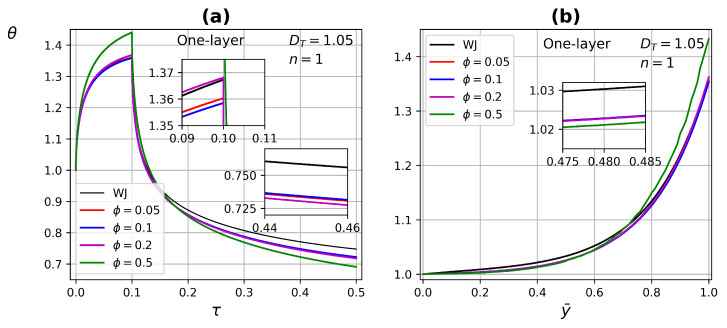
Temperature profiles as a function of (**a**) time and (**b**) tissue layer depth, for different porosity ϕ values at τ=0.09.

**Figure 13 micromachines-13-00424-f013:**
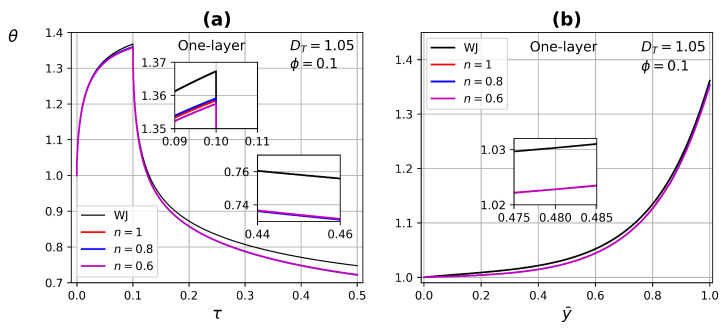
Temperature profiles as a function of (**a**) time and (**b**) tissue layer depth for different power-law index *n* at τ=0.09 and porosity ϕ=0.1.

**Figure 14 micromachines-13-00424-f014:**
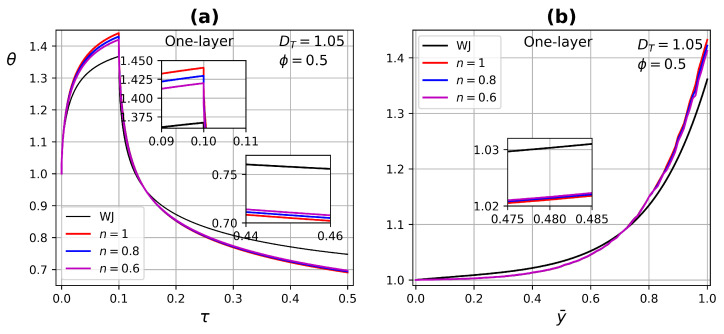
Temperature profiles as a function of (**a**) time and (**b**) tissue layer depth, for different power-law index *n* at τ=0.09 and porosity ϕ=0.5.

**Figure 15 micromachines-13-00424-f015:**
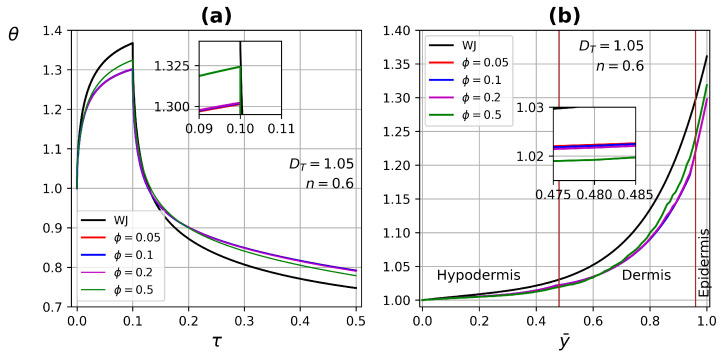
Temperature profiles as a function of (**a**) time and (**b**) tissue three-layer depth with different porosities, ϕ=0, 0.5, and 0.05 for epidermis, dermis, and hypodermis, respectively.

**Figure 16 micromachines-13-00424-f016:**
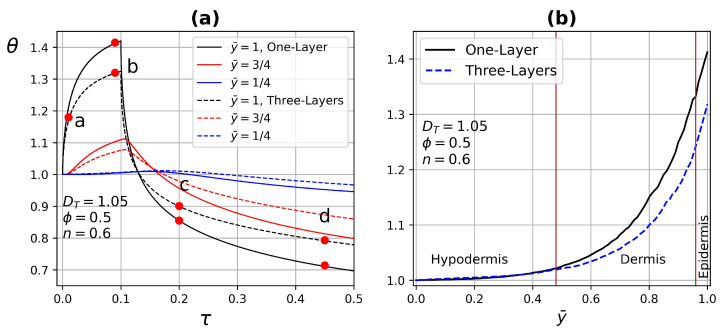
Temperature profile comparison between one- and three-layer tissue as a function of (**a**) time at three different depths of the tissue and (**b**) depth of the three-layer tissue with different porosities ϕ=0, 0.5, and 0.05 for epidermis, dermis, and hypodermis, respectively.

**Figure 17 micromachines-13-00424-f017:**
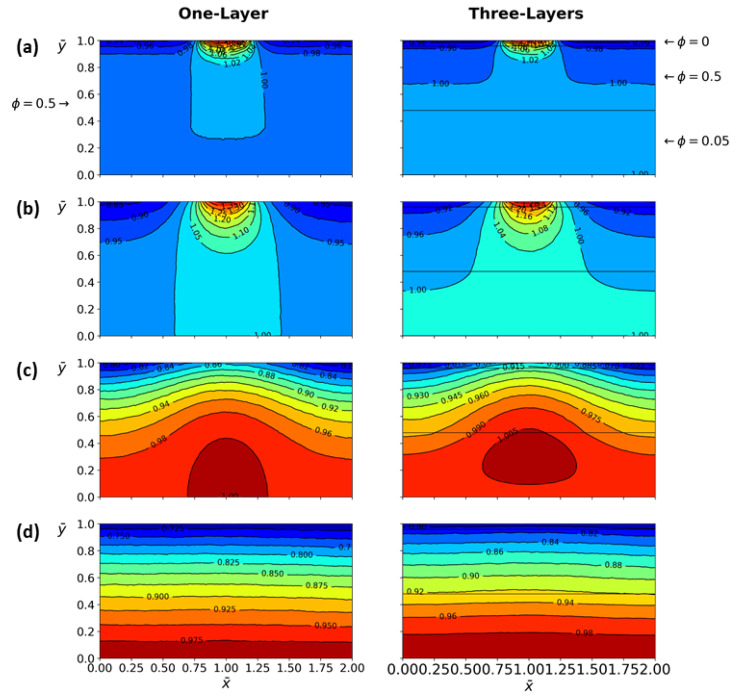
Comparison of the temperature distribution for one-layer tissue with ϕ=0.5 and three-layer tissue with ϕ=0, 0.5, and 0.05 for epidermis, dermis, and hypodermis, respectively. For different times from top to bottom (**a**) τ=0.01, (**b**) τ=0.09, (**c**) τ=0.2, and (**d**) τ=0.45. For a non-Newtonian fluid, n=0.6.

**Figure 18 micromachines-13-00424-f018:**
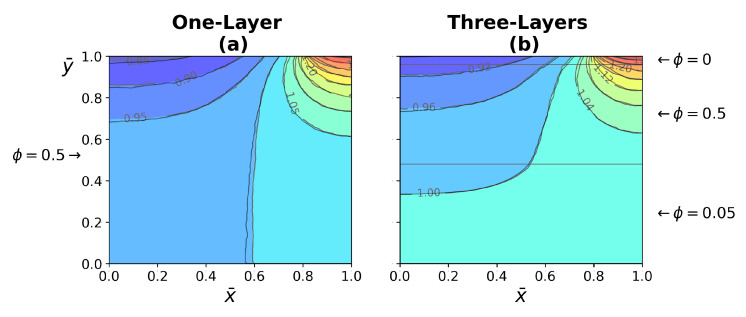
The right half is overlapped on the left half of the temperature distribution for the one- and three-layer tissue. Temperature distribution was taken from Figure 17 at time τ=0.09.

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
