# Peer review of "Calculation of Effective Thermal Conductivity for Human Skin Using the Fractal Monte Carlo Method"

_micromachines, 2022, doi:10.3390/mi13030424_

Round 1

Reviewer 1 Report (Previous Reviewer 1)

Comments and Suggestions for Authors

In this revised manuscript, the authors made significant improvement according to my comments. But some minor revisions should be further made before this manuscript can be accepted for publication. Here are my detailed comments.

  1.  The citation of Eq. (18) was incomplete because the method and model were first proposed by Wu and Yu in 2007, and in fact Eq. (18) is the same as that derived by Wu and Yu, see Eq. (12) in Wu and Yu’s paper (J. Wu and B. Yu, A fractal resistance model for flow through porous media, Int. J. Heat Mass Transfer 50, 3925-3932 (2007)). Citation should be given to a theory/model/method founder(s) as the earliest as possible.
  2.  The citation of Eq. (27) was missing, and the citation should be given to Ref. [50], and citation should be provided immediately before citing it.
  3. After line 179, the authors should add a brief discussion: Feng et al. [?] compared the ratios of λminmax in Eq. (27) with the experimental data (see Fig. 2 in Feng et al.’ paper, A generalized model for the effective thermal conductivity of porous media based on self-similarity, J. Phys. D. Applied Physics, 37, 3030-3040(2004)) reported in literature and found that the ratio of λminmax ≅ 10-3 is more suitable to most data.

Author Response

Reviewer 2 Report (Previous Reviewer 2)

This review report has been removed from the review record as it did not meet MDPI’s review report standards (https://www.mdpi.com/reviewers#_bookmark11).

Reviewer 3 Report (New Reviewer)

Comments and Suggestions for Authors

I proceeded to analyze the manuscript entitled:

Calculation of Effective Thermal Conductivity for Human Skin Using the Fractal Monte Carlo Method, written by:

Guillermo Rojas-Altamirano, René O. Vargas, Juan P. Escandón, Rubén Mil-Martínez and Alan Rojas-Montero

This manuscript present a numerical model to estimate the effective thermal conductivity (ETC) of human skin by treating it as a porous medium. The approach combines fractal scaling laws with Monte Carlo simulations to represent the vascular pore structure. Blood is modeled as a shear-thinning non-Newtonian fluid and the simulations are performed for a Newtonian fluid, as well. The model simulates heat transfer in both single- and three-layer skin configurations under surface heating conditions (as in hyperthermia treatment) and the results are compared with the Weinbaum and Jiji model.

The article illustrates a substantial amount of theoretical and computational effort.

The topic is, in my opinion, quite narrow but might be interesting for researchers interested in heat transfer in biological structures. The figures are suggestive and in many details and support the statements.

References are in decent amount and reveal that authors are well aware of what was written in the field. The article is well written, using good English, in my opinion. There are several phrases that are a bit confusing, yet not many.

Yet I found aspects that, in my opinion, raise concerns and require improvement and additional clarification, as indicated on each item, and they are mentioned below.

Minor concerns:

-eq. (11) is incorrect. Verify and correct.

- 4. Results: “The numerical results presented in this section were generated from a numerical

code developed in the Fortran programming language.” Clarify whether you wrote the code or you used a code, in which case give details. What Fortran version?

-verify the captions of Figs 11, 12 13 and 14: the abscissa of the a) subplot is tau, it cannot be anything at tau=0.9

Major concerns:

1. At the end of Introduction: “To the authors’ knowledge, there are no studies

that determine an ETC for heat transfer in biological tissues, considering the techniques

mentioned above and especially for non-Newtonian fluids.”

The actual contribution here is the specific combination of fractal + Monte Carlo methods, not ETC for heat transfer in biological tissues. There are papers written on this topic, as your references [39] and [32], which should be mentioned at this statement and the claim adjusted accordingly.

2. The thermal conductivity, perfusion rate, and viscosity are held constant despite known dependence on temperature and physiological response (e.g., vasodilation during heating). The authors should comment on the errors introduced by these approximations.

3. The model is compared with the predictions of the Weinbaum–Jiji model, which itself is theoretical, not with experimental data. This is a limitation in validating the model. A comparison of the predictions with ETC reports in literature would strengthen the manuscript.

4. The Monte Carlo Simulation is not detailed. Details as the number of samples, a discussion of the number of samples with respect to convergence of results are not presented, therefore it is difficult to reproduce and verify the work. More details should be given in manuscript.

Round 2

Reviewer 2 Report (Previous Reviewer 2)

This review report has been removed from the review record as it did not meet MDPI’s review report standards (https://www.mdpi.com/reviewers#_bookmark11).

This manuscript is a resubmission of an earlier submission. The following is a list of the peer review reports and author responses from that submission.

Round 1

Reviewer 1 Report

Comments and Suggestions for Authors

In the present study, a mathematical model is developed to determine the effective thermal conductivity of the human skin based on the Fractal-Monte Carlo Method. Despite the fact the manuscript is a well-written and focused paper, the novelty and utilized assumptions are the major issues. As the solved governing equation is a simplified 2D heat transfer equation, the main focus of this paper is on the estimation of the effective conductivity. The utilization of fractal scaling and Monte Carlo methods is greatly out of date for the current problem and several more novel methods are implemented to achieve better compatibility with the nature of the problem. For example, the presence of tissue porosity and tortuosity, as well as capillaries can’t be modeled by a single conductivity parameter in the bioheat equation, especially considering more sophisticated recent publications. Recent studies apply these effects on the entire structure of the governing equation using fractional calculus (not only on a single parameter). Besides, the temperature of blood and tissue are assumed equal in this study using local thermal equilibrium assumption which is not acceptable for an investigation focused merely on skin thermal conductivity.  

Reviewer 2 Report

Comments and Suggestions for Authors

In this work, the authors applied the Fractal-Monte Carlo method to describe the tissue as a porous medium, the blood was considered as Newtonian and non-Newtonian fluid for comparative and analytical purposes. The effective thermal conductivity (ETC) for living tissues was determined. The effect of the principal variables such as fractal dimensions DT and Df , porosity and the power-law index on the temperature profiles were analyzed. Although this type of research is interesting, this manuscript has some severe errors, and this manuscript should be rejected for publication. However, if the authors are willing to make substantial revisions according to my comments, I would be glad to re-review this revised manuscript. Here my detailed comments.

Main comments:
1. Eq. (25) lacks both physical and mathematical bases because Ri in Eq. (25) are the random numbers uniformly distributed between 0-1, thus λi is uniformly distributed between λminmax. Therefore, the capillary diameter λi produced by Eq. (25) is contradictory or inconsistent with that defined by Eq. (9), which indicates that the diameter/size λi distribution of capillaries also depends on the fractal dimension Df, and thus Eq. (9) indicates that the diameter/size λi of capillaries in a set of fractal capillaries is non-uniformly distributed between λminmax. While, Eq. (25) indicates that the diameter/size λi is uniformly distributed between λminmax since Ri in Eq. (25) is the random number uniformly distributed between 0-1. Therefore, λi determined by Eq. (25) is nothing to do with fractal.

2. From the results shown in Fig. 2 (c) and (d), I find that there are many maximum pore sizes with almost the same sizes. This is contradictory/inconsistent with Eq. (9) because Eq. (9) implies that there is only one maximum pore in a sent of fractal pores or fractal objects, see the review article by Yu (B. Yu, Analysis of flow in fractal porous media, Appl. Mech. Rev. 61(5), 050801(2008)). I suggest that the authors consult the reports/figures published in many papers, see the comment 4# below for recommended papers.

3. Since Eq. (25) lacks both physical and mathematical bases, all relevant computations such as Eqs. (18), (20) and (27) calculated on the basis of Eq. (25) were unconvinced. All figures based on Eq. (25) should be redrawn.

4. Above Eq. (25), the sentence: “This method has been used in similar work such as Yu et al. [46]” should be revised as: “Yu et al [46] was the first to propose the Fractal-Monte Carlo methodology to simulate the transports in fractal porous media”. Then the authors should make a brief review on the research progress in area of Fractal-Monte Carlo methodology. Here I am recommending some references for authors’ reference (in the order of published years):
1). M. Zou, B. Yu, Y. Feng, P. Xu, A Monte Carlo method for simulating fractal surfaces, Physica A 386 (2007) 176-186.
2). Y. Feng, B. Yu, P. Xu and M. Zou, Thermal conductivity of nanofluids and size distribution of nanoparticles by Monte Carlo simulations, J. Nanopart. Res. 10(2008)1319-1328.
3). P. Xu, B. Yu, X. Qiao, S. Qiu, Z. Jiang, Radial permeability of fractured porous media by Monte Carlo simulations, Int. J. Heat Mass Transfer 57 (2013) 369–374.
4). U. Vadapalli, R.P. Srivastava, N. Vedanti, Estimation of permeability of a sandstone reservoir by a fractal and Monte Carlo simulation approach: a case study, Nonlinear Proc. Geoph. 21 (2014) 9-18.
5). Y. Xu, Y. Zheng, J. Kou, Prediction of effective thermal conductivity of porous media with fractal-Monte Carlo simulations, Fractals, 22(3) (2014) 1440004.
6). B. Xiao, X. Zhang, G. Jiang, G. Long, G. Liu, Kozeny-Carman Constant for Gas Flow Through Fibrous Porous Media by Fractal-Monte Carlo Simulations, Fractals 27(4) (2019) 1950062.
7). H. Su, Y. Zhang, B. Xiao, X. Huang, B. Yu, A Fractal-Monte Carlo approach to model oil and water two-phase seepage in low-permeability reservoirs with fractal rough surfaces, Fractals 29(1) (2021)2150003
8). J. Yang, M. Wang, L. Wu, Y. Liu, S. Qiu, P. Xu, A novel Monte Carlo simulation on gas flow in fractal shale reservoir, Energy 236 (2021)121513.

In all, based on my above comments, I found that there were some critical errors, and thus this manuscript should be rejected for publication. However, if the authors are willing to make substantial revisions according to my above comments, I would be glad to re-review this revised version.

Minor comments:
1. The citation of Eq. (21) should be provided immediately before citing it and should be given to Ref. [41] because citation should be given to a theory/model/equation founder as the earliest as possible.